# The Multilevel Mechanism of Multifoci Service Orientation on Emotional Labor: Based on the Chinese Hospitality Industry

**DOI:** 10.3390/ijerph17124314

**Published:** 2020-06-17

**Authors:** Yong Yang, Fan Yang, Jingzhu Cao, Bo Feng

**Affiliations:** 1School of Business Administration, Northeastern University, Shenyang 110167, China; 1901231@stu.neu.edu.cn (F.Y.); 1901926@stu.neu.edu.cn (J.C.); 1901928@stu.neu.edu.cn (B.F.); 2School of Management, Northeastern University at Qinhuangdao, Qinhuangdao 066004, China

**Keywords:** multifoci service orientation, multifoci social capital, emotional labor, multilevel analysis

## Abstract

Emotional labor exerts a significant impact on employees’ physical and mental health as a unique form of labor. This study aims to explore the multilevel mechanism of multifoci service orientation on emotional labor from the social capital perspective. Through a multistage survey of managers and employees of 31 hospitality service companies in China, we obtained a valid sample of 31 business managers and 760 employees from hotel, restaurant, and tourism. Using Statistical analytical tools, the results revealed that organizational service orientation and individual service orientation enhanced individual social capital, which promoted employees’ deep acting and exerted a partial mediation effect on the correlation between individual service orientation and deep acting. Besides, individual social capital exerted a partial mediation effect on the relationship between organizational service orientation and deep acting. At the organizational level, organizational service orientation positively affected organizational social capital; and organizational social capital positively affected aggregate deep acting. The study results provide theoretical guidance for service companies to enhance employees’ well-being.

## 1. Introduction

The hospitality service industry includes hotels, restaurants, health services, tourism, and transportation, which exerts a crucial impact on people’s quality of life and happiness. The biggest opportunity for hospitality companies to gain a competitive advantage is whether they could offer customers high-quality hospitality services [1]. Capiel et al. reported that up to 67% of customers stopped patronizing the company because of the indifferent attitude of service employees [2]. Nevertheless, as service employees serve many customers every day, it is challenging to make them smile all the time. When the inner feelings of service employees do not meet the requirements of their company (e.g., anger and frustration), they have to make considerable efforts to adjust their negative emotions to express the enthusiasm and concern; This process of emotional regulation is termed as “emotional labor” [3,4]. Emotional labor includes two dimensions: surface acting and deep acting [5,6,7]. Surface acting implies employees are pretending to be happy but still feeling negative, while deep acting implies employees spontaneously changing their negative inner feelings to adapt to the external environment. To date, numerous studies have demonstrated that surface acting could cause emotional exhaustion and work dissatisfaction [8,9,10,11]. By contrast, deep acting could markedly improve employees’ job accomplishment and provide good service experience for customers [10,12]. Thus, how to suppress surface acting and promote the deep acting of employees has become one of the major problems experienced by service managers.

In order to reduce the risk of physical and mental health of service employees, recent studies have paid special attention to how to inhibit surface acting and promote deep acting. The existing studies mainly examined the antecedents of emotional labor from the organizational environment and individual trait. Individual trait factors include big five personality [13], negative personality [14], emotional intelligence [15,16], self-monitoring [17], and self-determined motivation [18,19]; Organizational environment includes authentic leadership [20], social and organizational support [21,22], service climate [23], service leadership [24], supervisor support [25], organizational justice [26], and customer misbehavior [27].

Although scholars in the field of management and service have made some progress in discussing the topic, the following issues persist: first, the existing literature primarily focuses on single-level factors. Kozlowski and Klein argued that organizations are multilevel systems and organizational macro-environmental factors may affect the micro-level employee attitudes and behaviors [28]. According to Kozlowski and Klein’s view, the multilevel theoretical perspective is combining macro-environmental and micro individual trait factors, and discussing how they have an impact on employees’ attitudes and behaviors commonly. Based on the multilevel theoretical perspective [29], if only one level of antecedent factors of emotional labor is focused at, another level of meaningful explanation of emotional labor could be missed; It is also impossible to explain how organizational context and individual trait conjointly affect emotional labor. Besides, at the organizational level of a company, Ashkanasy et al. believed that the emotional expression of employees could be aggregated to the organizational level and become an organizational emotional climate to enhance all the employees’ wellbeing through recruitment, selection, training, and daily service encounter management [30,31]. However, their claims have not been verified, which is also an interesting question that needs further discussion. Lastly, service orientation and social capital are recognized as multifoci constructs. Multifoci service orientation refers to the different effects of service orientation types according to their source. Existing literature conceptualize service orientation from two sources, i.e., a service employee and a service organization [32,33]. The former is named by individual orientation refers to service employees’ intention to providing quality service [32]; the latter is named by organizational service orientation refers to service organization’s intention to providing quality service [33]. Similar to service orientation, Social capital is also a multifoci conception that has two sources: a service employee and a service organization [34,35], which is named by individual social capital and organizational social capital respectively. Individual social capital refers to resources (e.g., material resources, information, and trust) embedded in the employee’s social network (including employee’s relationship with colleagues, superiors, and customers) [34]; Organizational social capital refers to resources embedded in the organization’s social network, including structural (the connection between members), relational (trust between members) and cognitive (common goals between members) resources [35]. This is important because employees typically experience related yet distinct service orientation and social capital at the organizational and individual sources. More specifically, a multi-foci approach can identify which particular service orientation and social capital types and sources are most influential in affecting employee emotional labor.

Hence, this study aims to integrate multifoci service orientation and multifoci social capital and discuss how they affect emotional labor (both organizational and individual level). The remainder of this paper is arranged as follows: firstly, we propose the hypotheses of the effect of organizational service orientation, individual service orientation, organizational social capital, individual social capital on emotional labor; secondly, the research design is presented (including data collection and conception measurement); thirdly, data analysis and discussion are provided; finally, conclusions, management implication, and limitations and future research directions are provided.

## 2. Literature Review

### Social Capital

Social capital has received strong attention from sociologists since the end of the 20th century. Several researchers made important contributions to the development of social capital theory, such as Bourdieu [36], Coleman [37], Putnam [38,39], and Lin [40]. Their theoretical views anchored the concept in the social sciences. There are two perspectives of social capital in the field of social sciences with society-focused social capital and individual-focused social capital. The former representative scholar is Putnam. In his books “Making Democracy Work: Civic Traditions in Modern Italy” [38] and “Bowling Alone: The Collapse and Revival of American Community” [39], he focuses on the impact of social capital elements (such as citizen participation, sharing norms and social trust) on institutional performance, economic prosperity and social development; the latter representative scholar is Lin. In his book “Social Capital: A Theory of Social Structure and Action” [40], he focuses on how social individuals interact with other actors and obtain resources from other actors to get better returns driven by instrumental needs or expressive needs. Since then, scholars have developed the definition of individual-focused social capital [41] and group-focused social capital [39]. Individual-focused social capital emphasizes that individuals obtain the right to use resources (e.g., material resources, information, and emotional support) through their social network. Individuals need to continuously manage their relationships to maintain their social capital and use resources required [42]. Group-focused social capital takes macro-level social structure (such as an organization) as the basic analysis unit. Researchers in this perspective focuses on the relationship and quality of actors in the network structure, and believe that social capital is like an invisible resource (e.g., trust and rules) embedded in the social structure [43]. Group-focused social capital can promote collective cohesion and the cooperative behavior of group members [42]. Although researchers pay attention to the different perspectives of social capital, they have a certain consensus on the definition of social capital. They regard social capital as resources embedded in social networks and essential power for driving individual and collective behavior [44].

Management researchers integrated the group-focused and individual-focused social capital perspective in the social sciences (for example, Adler and Kwon [42]; Nahapiet and Ghoshal [34]). Nahapiet and Ghoshal [34] proposed that social capital is a multi-dimensional construct, which includes structural, relationship, and cognitive dimensions. Structural social capital refers to an organizational member striving to develop an interpersonal relationship with others; cognitive social capital refers to the goals, meanings and values shared among organizational members; relational social capital refers to the quality and content of the relationship between organizational members, such as trust, norms, and obligations. The three dimensions of social capital can exist either in the social network of an organization or in the social network of an individual [35]. Leana and Pil divided social capital into two types according to the type of organizational member relationship [45]: internal social capital and external social capital. Internal social capital refers to the structural, cognitive, and relational dimensions of social capital arising from the relationships between members within an organization. External social capital refers to the structural, cognitive, and relational dimensions of social capital resulting from the relationship between the organization and external stakeholders. In recent years, management social capital researchers mainly discussed the impact of social capital on organizational performance, employee performance, and innovation performance [46,47]. Only several literatures discuss the causes of social capital, For example, self-sacrificial leadership [48], club’s social events involvement [47], human resource management practices [49].

## 3. Hypotheses Development

### 3.1. The Influence of Multifoci Service Orientation on Individual Social Capital

Individual service orientation refers to “a service employee’s intention to provide quality service” [32]. Arasli et al. argued employees with high service orientation are more likely to display traits like enjoying helping others, being considerate and patient, and being warm [50]; they are particularly suitable for interpersonal communication and are easier to gain the trust of colleagues or customers. Once service employees build trustful relationships with colleagues or customers, they are easier to obtain the resources from colleagues and customers, for example, emotional support [51]. These resources are embedded in employees’ social networks, i.e., social capital. For example, Lin points out individual social capital is the resources individual obtained through interaction with others [40]. Hence, we propose the following hypothesis:

**Hypothesis 1** **(H1):**
*Individual service orientation exerts a positive impact on individual social capital.*


Lytle and Timmerman validated that organizational service orientation supports interpersonal communication among organizational members through the following four aspects: service-oriented leadership; service encounter management; service technology system; and human resource training [33]. A service organization with high service orientation offers favorable environmental conditions for the interaction among service employees, leaders, colleagues, and customers because its culture focuses on “providing high-quality services to customers”. individual social capital denotes service employee obtain resources from his/her social network [45]. When his/her organization provide more chances for building a relationship with others, he/she could use more resources embedded in his/her social relationships; concretely, the training program improves service employees’ communication skills; managers who use service leadership respect service employees and spend more time on communicating with employees and sharing their experience and information resources with then; service technology system changes the way employees communicate with customers and colleagues. For instance, employees use mobile social software to expand their social network; service encounter management supports highly efficient interaction between employees, colleagues, and customers. Indeed, several studies have demonstrated that supportive leadership and a trusting organizational environment could enhance employees’ social capital [52,53]. Ellinger et al. found that if managers spend more time with employees, encourage and respect service employees, they will improve the social resources of employees [52]. Hence, we propose the following hypotheses:

**Hypothesis 2** **(H2):***Organizational service orientation exerts a positive impact on individual social capital*.

### 3.2. The Influence of Individual Social Capital on Emotional Labor

The main part of the service industry is offering delight service for customers [54]. Service organizations often require frontline service employees to express enthusiasm and friendliness to customers. When a service employee’s inner feeling is not positive, he/she must express happiness through emotional labor. Emotional labor includes two dimensions: surface acting and deep acting [3]. Surface acting is that an employee expresses happiness and enthusiasm to customers by suppressing their inner anger, depression, or other negative emotions [10]. Deep acting implies employees change their inner negative feelings to present themselves full of happiness and enthusiasm [10,11]. Researchers on emotional labor verified service employees need resources to regulate their negative feelings [10,11]. Social capital as an employee’s social resource (e.g., emotional support from colleagues) can make up for the employee’s effort to regulate emotion [45]. So, a service employee with high social capital can adjust his/her negative emotions and eliminate them as soon as possible with the help of colleagues. That is, a service employee with high social capital uses deep acting to serve customers rather than surface acting that he/she retains negative feelings. Xu et al. used the meta-analysis method to examine emotional labor’s antecedents and found that social support could inhibit surface acting and promote deep acting [21]. Accordingly, the following hypothesis is proposed:

**Hypothesis 3** **(H3):***Individual social capital exerts a negative impact on surface acting*.

**Hypothesis 4** **(H4):***Individual social capital exerts a positive impact on deep acting*.

### 3.3. Individual Social Capital Exerts a Mediating Effect on the Relationship between Individual Service Orientation and Emotional Labor

Based on the hierarchical model of personality theory [55], service orientation belongs to employees’ traits in the service context, which is shaped by their inherent personality and service work. Service orientation is a frontline employee’s predisposition to attain high service performance. Social capital denotes an employee’s interpersonal resources accrued in the daily service work [45]. Employees who have considerate, friendly, humble, and enthusiastic are more likely to win the trust of leaders, colleagues, and customers, which could help employees to regulate negative emotions. For example, employees’ colleagues and immediate superior could help them get rid of negative moods and be happy as soon as possible, as well as help them inhibit surface acting that harms their physical and mental health and promote deep acting that is conducive to enhancing service performance. Hence, we predict that individual social capital plays a mediating role between individual service orientation and emotional labor. The hypotheses are as follows:

**Hypothesis 5** **(H5):***Individual social capital exerts a mediating effect on the relationship between individual service orientation and surface acting*.

**Hypothesis 6** **(H6):***Individual social capital exerts a mediating effect on the relationship between individual service orientation and deep acting*.

### 3.4. Individual Social Capital Exerts a Mediating Effect on the Relationship between Organizational Service Orientation and Emotional Labor

Resource conservation theory argues that individual psychological resources are limited, and they have the basic motivation to preserve and protect their resources from loss [56]. Resource conservation theory postulates that when employees pay psychological resources for their work, they might face the risk of personal resources loss without external resources supplement, which could lead to employees’ enthusiasm decline and burnout. In turn, employees tend to decrease the cost of subsequent resources. Employees can continue to pay for their psychological resources only with the supplement of external resources. Emotional labor requires employees to adjust their emotions in their work and express their enthusiasm and happiness; this emotional regulation process needs the consumption of more psychological resources [3]. As an individual resource, social capital can supplement the resource depletion of service employees engaged in emotional labor. Moreover, organizational service orientation could provide an external favorable environment for employees to accumulate more social capital resources in four aspects, including managers, human resource training, empowerment, and service technology system support [57,58]. Hence, the following hypotheses were proposed:

**Hypothesis 7** **(H7):***Individual social capital exerts a mediating effect on the relationship between organizational service orientation and surface acting*.

**Hypothesis 8** **(H8):***Individual social capital exerts a mediating effect on the relationship between organizational service orientation and deep acting*.

### 3.5. The Impact of Organizational Service Orientation on Organizational Social Capital

Organization and management researchers have proposed two types of social capital, organizational social capital and individual social capital [59], which denotes that organizational actors accumulate social resources at organizational and individual levels, respectively. The former is called organizational social capital that implies “a resource reflecting the characteristics of social relations within an organization.” Organizational social capital can be considered an asset that can exert a positive impact on the organization itself and employees in the organization [45]. Reportedly, organizational social capital has three dimensions—common goals, trust, and interactional norms [60]. Organizational service orientation is the service organizations’ strategy and organizational culture. In the high service-oriented organization, managers need to unite employees’ common goals and enhance the level of trust between employees and customers, colleagues, and direct supervisors [61]. The cohesion of common goals and high-level trust is conducive to the formation of organizational social capital. Accordingly, we proposed the following hypothesis:

**Hypothesis 9** **(H9):***Organizational service orientation exerts a positive impact on organizational social capital*.

### 3.6. The Impact of Organizational Social Capital on Emotional Labor

Individual emotional labor could exist at the organizational level for the following reason: the service experience of customers is determined by all the employees they come in contact with. For example, customers’ dining experience in a restaurant is decided by the reception, catering attendant, cashier, and so on [2]. Employees at an organization could adopt similar emotional expressions through organizational recruitment, training, and emotional expression rules. Thus, emotional labor at the individual level could be up to the organizational level and become a phenomenon that depicts the overall emotional expressive behavior of employees at the organizational level. Furthermore, Ashakanasy et al. argued that emotional labor exists in enterprises as an organizational emotional atmosphere phenomenon [30,31].

Just as individual-level social capital promotes deep acting and constrains surface acting, organizational social capital could exert a positive impact on deep acting and inhibit surface acting at the organizational level. Suseno and Rowley highlighted that the harmonious climate among organizational members could enhance employees’ perception of organizational vision, which effectively improves employees’ emotional expression [53]. Precisely, in a harmonious climate, an employee trusts his/her colleague and the immediate supervisor and could get their help when he/she encounters annoying problems. The employee has sufficient emotional support to alleviate negative emotions. Conversely, if too little trustful climate exists between organizational members, the employee who needs colleagues’ or managers’ help might find it difficult to handle tough customers or other annoying problems. Furthermore, he/she could find it difficult to obtain the support of external resources; that is, the employee only chooses surface acting that retains inner negative feelings. Hence, the following hypotheses were proposed:

**Hypothesis 10** **(H10):***Organization social capital exerts a negative impact on aggregated surface acting*.

**Hypothesis 11** **(H11):***Organization social capital exerts a positive impact on aggregated surface acting*.

## 4. Research Design

### 4.1. Data Collection

We collected data by multistage and multisource method, which can effectively decrease common method bias. In addition, we trained the questionnaire distribution personnel in advance to ensure the quality of data collection. We asked the questionnaire distribution personnel to explain to the respondents (managers and employees) that the questionnaire was only used for academic research and the sought information was absolutely confidential. After filling in the questionnaire, the respondents sealed the questionnaire. In the entire process of investigation, the questionnaire distribution personnel did not contact the questionnaire, which is conducive to the privacy of questionnaire information.

The data collection was divided into three stages with a 1-month interval between each stage. In the first stage, we investigated 35 business department managers and 980 employees in 35 hospitality enterprises. The information provided by managers included organizational service orientation and organization information, while the information provided by employees included individual service orientation and personal information. At this stage, we obtained effective information of 33 business managers and 870 employees. The effective information rate of managers and employees was 94.29% and 88.78%, respectively. In the second stage, we examined the employees who obtained effective information in the first stage. The investigated information included individual social capital. At this stage, we obtained 790 employees’ effective information from 31 enterprises; the effective information rate of employees was 90.80%. In the third stage, we examined employees who obtained effective information in the second stage, including emotional labor. At this stage, we finally obtained 760 employees’ effective information from 31 enterprises; the effective information rate of employees was 96.20%. We observed no significant difference between the effective samples of 31 business managers and 760 employees obtained in the third stage and those obtained in the first stage (33 managers and 870 employees). Table 1 presents the final valid sample information.

### 4.2. Measurement

In this study, all variable measurement scales were obtained from English literature. We used the Brislin recommended method to adapt the English scale to Chinese versions [62]. The translation process was as follows: we invite two doctoral students who were proficient in Chinese and English; one doctoral student translated the English scale into Chinese, and the other doctoral student translated the Chinese version back into English. A working group comprised 1 author, 2 service management professors, and 2 service managers to discuss the differences between the original English scale and the Chinese version and the English scale that a doctoral student translated the Chinese version into. After four discussions (1 or 2 h per time), all members of the working group reached an agreement. Then, we let 7 middle managers and 70 service employees from 3 catering enterprises complete the questionnaire. The feedback of those managers and employees was good, which effectively guaranteed the content validity of our study variables’ scale. All the variables covered in the questionnaire were measured by the 7-point Likert scale where “1 represents total disagreement, and 7 represents total agreement.”

Individual-level variables included individual service orientation, individual social capital, emotional labor, and individual statistical variables.

#### 4.2.1. Individual Service Orientation

The scale of individual service orientation was modified from Hogan et al. with five items [32]. The typical topic was “I am willing to help others,” “It’s my job to meet the needs of customers.”

#### 4.2.2. Individual Social Capital

The scale of individual social capital was modified from Leana and Pil’s scale with 17 items [45]. The typical topic was “I will share my work problems with my colleagues,” “in general, my colleagues are trustworthy,” and “the purpose of my work is to achieve the goals of the company/department.”

#### 4.2.3. Emotional Labor

The scale of emotional labor was modified from Brotheridge and Grandey [10]. The scale includes two dimensions—surface acting and deep acting—with 8 items in total. The scale of surface acting includes 5 items, and the typical topic was “I often pretend to be happy and enthusiastic to customers.” The scale of deep acting includes 3 items, and the typical topic was “when facing customers, I will take the initiative to feel the emotion required by the company, not just changing the expression.”

Individual-level control variables included gender, marriage, age, income, education, and working years.

Organizational-level scale included organizational service orientation and control variables.

#### 4.2.4. Organizational Service Orientation

Organizational service orientation was modified from Lytle and Timmerman scale with 35 items [33]. The typical topics were as follows: “the company’s employees will minimize the inconvenience to customers,” “our company employees have great independent rights to decide how to provide high-quality services,” “our company uses advanced technology to support the work of frontline service employees,” and “we will actively listen to the opinions of customers.”

#### 4.2.5. Organize Social Capital

The average value of individual social capital was estimated to measure organizational social capital. The median of individual social capital RWG (J) was 0.93 (>0.7), ICC (1) was 0.21 (>0.12), and ICC (2) was 0.86 (> 0.7), which fulfilled the aggregation condition, that is, social capital could be aggregated from individual level to organizational level.

#### 4.2.6. Aggregated Emotional Labor

By evaluating the average value of each company employees’ surface acting and deep acting, we measured aggregated surface acting and aggregated deep acting. Among them, the median of RWG (j) of surface acting was 0.85 (>0.7); its average of RWG (j) was 0.78. The RWG (j) of seven groups was <0 7; ICC (1) was 0.20 (>0.12) and ICC (2) was 0.85 (>0.7). The results indicated that individual-level surface acting could aggregate into organizational-level surface acting. In addition, the median RWG (j) of deep acting was 0.83 (>0.7); its average was 0.81. Moreover, the RWG (j) of five groups was <0 7; ICC (1) was 0.13 (>0.12) and ICC (2) was 0.78 (>0.7). The results indicated that individual-level deep acting could be aggregated into organization-level deep acting.

Organization-level control variables included enterprise type, industrial distribution, years of establishment, and firm size.

Control variables selection principles. According to the Becker recommendation on selecting control variables [63], the reasons for the selection of organizational and individual level control variables as follows: Firstly, we follow the control variables selected in the literature of emotional labor antecedent, for example, Hu et al. [27] and Scott et al. [17] choose age, gender, working years, and marriage as control variables. They believe that these variables may have a potential impact on emotional labor. Control variables selection at the organizational level, we mainly follow organizational service orientation’s consequence literature, such as Jung [57], Luk et al. [58], and Chang [64], Grissemann [65]. In their studies, hotel size, industry distribution, and years of the establishment are considered as control variables, which may have a potential impact on individual and organizational performance. Besides, China’s service industry has its specialties. First, China’s service enterprises include two types, state-owned enterprises, and private enterprises. There are significant differences in the organizational environment (including organizational culture, employee employment policy, and manager’s leadership) between the two types of enterprises, which have differences in the environment affecting employees’ emotional management behavior. The literature that explores the Chinese service context takes enterprise type as one of the control variables, for example, Liu and Yang [66]. Second, the frontline service employees in the Chinese service industry do not take customers’ tip as their income, like other industries, they have a fixed income, with the differences in the firm’s size and grade, their income has obvious differences. Third, Chinese education belongs to nine-year compulsory education. Many employees are engaged in service work after nine-year junior high school education. Besides, many large-scale service enterprises recruit undergraduate or graduate students to engage in service management. Before these employees become managers, the company requires them to practice at the grass-roots service work for more than one year. Due to the above reasons, the educational background of front-line service staff is very different. Therefore, the studies on employee work attitude and behavior in the Chinese service context take employee’s income and education as control variables, such as Chen et al. [67].

In this study, the data analysis tools used were SPSS21.0, AMOS21.0, HLM7.0. SPSS21.0 was used for reliability test, correlation analysis, and regression analysis to test the correlation between organizational-level variables. In addition, AMOS21.0 was used for confirmatory factor analysis (CFA), while HLM7.0 was used to test the correlation between the cross-level variables. Before using HLM7.0 for data analysis, individual social capital, surface acting, and deep acting were taken as dependent variables. Besides, levels 1 and 2 were operated without any independent variables. A significant difference was noted between the variance within the individual social capital group and the variance between groups (*χ*² (31, *N* = 760) = 238.51, *p* < 0.001, ICC(1) = 20.80%), suggesting that the variance between groups of individual social capital could explain 20.80% of the total variance. In addition, we found significant differences between the variance within the surface acting group and the variance between groups (*χ*² (31, *N* = 760) = 227.48, *p* < 0.001, ICC(1) = 20.35%), which means that the variance between groups of surface acting could explain 20.35% of the total variance. Moreover, differences were noted between the variance within the deep acting group and the variance between groups (*χ*² (31, *N* = 760) = 151.35, *p* < 0.001, ICC(1) = 13.04%). Overall, the data results revealed that the variance between groups of surface acting could explain 13.04% of the total variance. Thus, the multilevel theoretical model of the study was suitable for the HLM7.0 test.

## 5. Data Analysis Results

### 5.1. Reliability, Validity, and Correlation Analysis

#### Reliability and Validity Test

*Reliability test*. The individual service orientation’s reliability was 0.883; individual social capital’s reliability was 0.954; surface acting’s reliability was 0.854; deep acting’s reliability was 0.756. In addition, organizational service orientation’s reliability was 0.956. The data results revealed that all variables in the study exhibited good reliability.

*Convergent validity test*. Table 2 and Table 3 present the factor loading of individual-level variables.

*Discriminate validity test*. We constructed measurement models for individual service orientation, individual social capital, and emotional labor (surface acting, deep acting; Table 4).

Table 4 shows that comparing the benchmark model with the other three measurement models revealed that the fitting validity of yjr benchmark model was the best. The benchmark model depicted the four-factor structure of individual service orientation, individual social capital, surface acting, and deep acting. It passed the discriminate validity test.

### 5.2. Correlation Analysis

We used the Pearson correlation analysis function of SPSS 21.0. Table 5 shows the descriptive statistics and the correlativity of the variables after controlling variables like gender, age, marital status, education, income, and working years.

### 5.3. Examining the Impact of Organizational Service Orientation and Individual Service Orientation on Individual Social Capital

Using HLM7 software, we performed a multilevel analysis with organizational service orientation and individual service orientation as the independent variables and individual social capital as the dependent variable. Table 6 presents the analysis results.

### 5.4. Examining the Impact of Individual Social Capital on Emotional Labor

Using HLM7 software, we performed regression analysis with individual service orientation as the independent variable and emotional labor as the dependent variable. Table 7 shows the analysis results.

### 5.5. Examining the Mediating Role of Individual Social Capital

We tested H5–H8 based on the mediating effect test procedure proposed by Preacher and Hayes [68] and Hayes and Preacher [69]. First, the regression coefficients of organizational service orientation, individual service orientation, and emotional labor are significant. Second, the regression coefficients of organizational service orientation, individual service orientation, and individual social capital are significant. Third, the regression coefficient of individual social capital and emotional labor is significant. Finally, we consider whether the regression coefficients of organizational service orientation, individual service orientation, and emotional labor were significant after joining individual social capital. If the regression coefficient was not significant, individual social capital played a mediation role completely; however, if the correlation was significant and the regression coefficient significantly decreased, individual social capital played a partial mediation role.

*The first condition test*. The total effect coefficient of individual service orientation on surface acting was 0.020 (*p* > 0.05, as shown in M1 of Table 8). The total effect coefficient of individual service orientation on deep acting was 0.300 (*p* < 0.001, as shown in M2 of Table 8). The total effect coefficient of organizational service orientation on surface acting was −0.147 (*p* > 0.05, as shown in M1 of Table 8). Furthermore, the total effect coefficient of organizational service orientation on deep acting was 0.094 (*p* < 0.01, as shown in M2 of Table 8).

*The second condition test*. The coefficient of individual service orientation on individual social capital was 0.501 (*p* < 0.001, as shown in M1 of Table 6). The coefficient of organizational service orientation on individual social capital was 0.141 (*p* < 0.01, as shown in M1 of Table 6).

*The third condition test*. The coefficients of individual social capital on surface acting and deep acting were 0.010 (*p* > 0.05) and 0.372 (*p* < 0.001), respectively (as shown in M1 and M2 of Table 7).

*The fourth condition test* (Table 9). After adding individual social capital variables, the direct effect coefficients of individual service orientation on surface acting and deep acting were −0.004 (*p* > 0.05) and 0.126 (*p* < 0.001), respectively. The direct effect coefficients of organizational service orientation on surface acting and deep acting were −0.160 (*p* > 0.05) and 0.065 (*p* < 0.05), respectively. Notably, H6 and H8 satisfied the abovementioned four conditions at the same time and were supported. That is, individual social capital plays a partial mediating role between individual service orientation and deep acting, as well as between organizational service orientation and deep acting.

### 5.6. Examining the Impact of Organizational Service Orientation and Organizational Social Capital

Table 10 shows the regression coefficient of organizational service orientation on organizational social capital after controlling variables of enterprise type, industrial distribution, years of establishment, and firm size.

### 5.7. Examining the Impact of Organizational Social Capital on Aggregated Emotional Labor

Table 11 shows the regression coefficient of organizational social capital on aggregated emotional labor after controlling variables of enterprise type, industrial distribution, years of establishment, and firm size.

The test results of all hypotheses in this study are shown in Table 12.

## 6. Discussion

First, most service industries maintain a competitive advantage by creating customer delight experience. Service employees’ emotional labor plays a crucial role in enhancing the positive emotions of customers. However, surface acting and deep acting exert different effects on employees’ physical and mental health. Lee and Madera systematically summarized the research progress of emotional labor literature in the field of the hospitality industry; they argued that limited studies had discussed multilevel factors of emotional labor [8]. On the other hand, several studies have discussed the impact of individual service orientation or customer orientation on employees’ emotional labor [26], which have neglected that service orientation theory exists in multiple levels of an organization. This study combines the service orientation theory from multiple perspectives to validate that both organizational service-oriented and individual service-oriented promote employees’ deep acting. The conclusion enriches the multilevel antecedents of emotional labor in the hospitality industry.

Second, emotional labor studies have primarily focused on the resource conservation theory and emotional event theory to explain the influencing factors of emotional labor [8]. For example, Hu et al. used the resource conservation theory to demonstrate the correlation between customer misconduct and emotional labor [27]. In addition, Lam and Chen and Hur et al. used the emotional event theory to illustrate the impact of organizational support on employees’ emotional state and emotional adjustment strategy [22,25]. Besides the resource conservation theory, we used the theory of trait hierarchy model to establish that individual social capital transforms external resources (organizational service orientation) and individual resources (individual service orientation) into deep acting, providing a new theoretical perspective for discussing the antecedents of emotional labor.

Finally, employees’ emotional expression is a crucial skill to achieve customer performance. At present, research in the service field primarily discusses individual-level research on emotional labor. Ashkanasy et al. believed that through recruitment, selection, training, and daily service encounter management, the emotional expression of employees could be aggregated at the organizational level and become an essential resource for a service organization to enhance customer performance [30,31]. In this study, we verified that individual-level emotional labor could be aggregated at the organizational level based on 31 hospitality firms’ data, and organizational service orientation promoted aggregated deep acting through organizational social capital. The research conclusion encourages more scholars to discuss emotional labor at the organizational level.

Although this study has received interesting theoretical implications, some hypotheses were not confirmed, which could be attributed to many reasons. First, individual social capital exerts no negative impact on surface acting, and their correlation is not supported at the organizational level (H3 and H10 are not supported), which could be because of the influence of Chinese cultural context; Chinese cultural tradition requires that individuals cover up their negative feelings without disturbing the feelings of others. Thus, some studies using Chinese employees in the service industry as survey samples approve that employees’ surface acting does not hurt customer performance or make customers feel insincere [70,71]. Instead, they believe that service employees do not bring negative feelings to work and obtain an understanding of customers and colleagues. From this standpoint, service employees in Chinese culture engaged in surface acting could bring customers’ understanding and colleagues’ support, and will not afford too many psychological resources, thereby not needing the supplement of organizational and individual social capital resources. Furthermore, individual social capital exerts no mediating effect between service orientation (organization and individual) and surface acting (H5 and H7 are not supported).

## 7. Conclusions

In sum, surface acting harms employee well-being and deep acting has a positive effect. The current study adds to the literature on antecedents of emotional labor (both individual level and organizational level) from multifoci service orientation and multifoci social capital is one of the underlying mechanisms through which this relationship occurs. Our findings suggest that deep acting at the individual level can be aggregated into an emotional climate at the organizational level, which is an important environmental factor to promote employee well-being and customer performance; employees with high service orientation have a warm and cheerful personality and like to help others. These traits are very suitable for interpersonal interaction and help employees to accumulate more social resources. And then, social resources come from customers and colleagues can promote deep acting and employees’ wellbeing, such as job accomplishment. Similarly, a service organization with high service orientation will establish a platform for employee experience exchange and provide support for all employees to improve emotional management skills. These strategies could help service employees to alleviate negative emotions in the work. Furthermore, a service-oriented organization has a high-quality social network (i.e., organizational social capital) among all the employees, which means employees work in a harmonious, trusting, and friendly organizational environment that will enhance all the employees’ wellbeing.

### 7.1. Management Implications

Service work is a long-time and high-intensity job and often suffers from unfair treatment by customers. Our findings can help service organization leaders to improve frontline service employees’ wellbeing in the following three aspects. First, for human resource managers, they should design a service-oriented personality test tool, and recruit personnel who have cheerful, optimistic, and like to help others. Such employees are suitable for communicating with customers and able to establish a harmonious working relationship with colleagues and leaders. And then, they could obtain emotional resources from them to regulate negative emotions and improve their workplace wellbeing. Secondly, for the senior leaders, they should cultivate a service-oriented climate through human resources management, service encounter management, service technical support, and service leadership. Service-oriented climate provides a large number of opportunities for service experience exchange among employees. service employees not only accumulate a lot of social resources to alleviate customer stress, but also have opportunities to improve emotional management skills. Thirdly, for junior managers, they should adopt service leadership to provide resourceful, emotional, and technical support for employees. Once employees encounter pressure from their families, colleagues, or customers, they need the help and support of immediate superior to adjust stressful situations with colleagues or customers. They are important forces for improving the mental state of employees.

### 7.2. Research Limitations and Future Research Directions

This study has obtained interesting conclusions but also has some limitations that should be improved in future research. First, we used many ways to decrease the common method deviation, such as multistage, multisource collection investigation data, and let the respondents fill in the questionnaire voluntarily and anonymously, but there remains some room for improvement. For example, we should let the employees’ supervisor assess their emotional labor. Second, we selected the enterprises in the hotel industry, catering industry, and the tourism industry as the research object, which might have limited our findings’ extension to other industries. In addition, the selected samples were mainly concentrated in northern China, which also limited our findings’ extension to the southern part of China. Finally, owing to the different cultural contexts between China and the West, there could be significant differences in emotional expressive norms followed by frontline employees. Hence, future research should further investigate the similarities and differences of emotional expressive norms between China and the West, as well as discuss its effect on the study theoretical model.

## Figures and Tables

**Table 1 ijerph-17-04314-t001:** Sample characteristics.

**Individual Level Sample Characteristics**							
**Demographic Variable**		**Number of Staff**	**Percentage**	**Demographic Variable**		**Number of Staff**	**Percentage**
Gender	Man	248	32.60%	Education Background	Below High School	151	19.90%
	Women	512	67.40%		High School	199	26.20%
Marriage	Married	415	54.60%		junior college	225	29.60%
	Single	345	45.40%		bachelor	148	19.50%
Age	Eighteen to twenty	53	7.00%		Master and above	37	4.80%
	Twenty-one to five	185	34.30%	Years of Working	Below one year	91	12.00%
	Twenty-six to thirty	286	37.70%		One to five	486	64.00%
	Thirty-one to five	151	19.90%		Six to ten	135	17.80%
	Thirty-six to forty	35	4.60%		Eleven to fifteen	25	3.30%
	Forty-one and above	50	6.50%		Fifteen and above	23	2.90%
Income (yuan)	Below 1500	56	7.40%	Income (yuan)	3501 to 4500	73	9.60%
	1501 to 2500	320	42.10%		4501 and above	38	5.00%
	2501 to 3500	244	32.10%				
**Organizational Level Sample Characteristics**							
**Statistical Variables**		**Number of Enterprises**	**Percentage**	**Statistical Variables**		**Number of Enterprises**	**Percentage**
Type of Enterprise	state-owned	11	35.50%	Industry Distribution	Hotel	8	25.80%
	Private	20	64.50%		Restaurant	15	48.40%
Years of Establishment (Year)	Below five year	5	16.10%		Tourism	8	25.80%
	Five to ten	10	32.30%	Firm Size (Number of employees)	Below fifty	7	22.60%
	Eleven to fifty	13	41.90%		Fifty to one hundred	7	22.60%
	Fifty and above	3	9.70%		Hundred-one to five hundred	12	38.70%
					Five hundred and above	5	16.10%

**Table 2 ijerph-17-04314-t002:** Individual service orientation and emotional labor’s convergence validity test.

Variable Name	Item	Factor Loading
Individual service orientation	I am willing to help others	0.793
	I can imagine that the best job is to help others solve difficulties	0.759
	I can get along with most people	0.814
	I’m proud of thoughtful service	0.817
	It’s my job to meet the needs of customers	0.714
Surfacing acting	when I face others at work, I often pretend to be happy and enthusiastic	0.592
	In order to better serve customers, I often cover up our true feelings	0.568
	When communicating with others at work, I seem to be acting	0.835
	I’m just pretending to express the emotions I need to work	0.824
	In order to express the emotions needed for work, I seem to put on some kind of “Mask” such as smiley face mask	0.810
Deep acting	when facing customers, I will take the initiative to feel the emotion expressed by the company	0.510
	When serving customers, I not only look happy and enthusiastic but also make my heart happy	0.875
	When serving customers, I will adjust my bad mood and let my enthusiasm come from Inner	0.815

**Table 3 ijerph-17-04314-t003:** Convergence validity test of individual social capital.

Variable Name	Item	Factor Loading
Individual social capital	The communication between me and my colleagues is open and sincere	0.756
	The work problems I encounter are shared with colleagues	0.767
	In my work, I accept other people’s friendly criticism and useful suggestions	0.793
	If a colleague affects work due to personal problems, I will communicate with her	0.758
	I am willing to share work experience and information with colleagues	0.793
	If I have a problem need to consult someone else, I can contact my colleagues at any time	0.805
	When I encounter difficulties, I can rely on colleagues to help solve	0.714
	At work, I usually consider the feelings of others	0.800
	My colleagues think me an honest person	0.736
	Overall, my colleagues are trustworthy	0.723
	I have the same ideals as my colleagues	0.682
	In order to achieve the collective goal and mission, I am full of energy	0.781
	My colleagues and I work with the same purpose	0.705
	The purpose of my work is to achieve the goal of the company / Department	0.769
	The company’s future development plan is the result of collective wisdom	0.757
	In the future company’s development, I agree with company leaders and colleagues	0.712

Table 2 and Table 3 show that the factor loading of individual service orientation, individual social capital, and emotional labor was >0.5, implying that all individual-level variables passed the convergence validity test.

**Table 4 ijerph-17-04314-t004:** Comparison of measurement models.

Indicator	χ^2^/df	GFI	TLI	NFI	IFI	CFI	RMSEA
Judgment criteria	<3	>0.9	>0.9	>0.9	>0.9	>0.9	<0.08
Benchmark model	2.975	0.917	0.958	0.938	0.958	0.948	0.051
Model 1	8.386	0.795	0.839	0.822	0.840	0.806	0.099
Model 2	6.907	0.831	0.872	0.854	0.872	0.845	0.088
Model 3	5.878	0.830	0.895	0.876	0.895	0.872	0.080

Note: Benchmark model: individual service orientation; individual social capital; surface acting; deep acting. Model 1: individual service orientation + individual social capital + surface performance + deep acting. Model 2: individual service orientation; individual social capital + surface acting + deep acting. Model 3: individual service orientation; individual social capital; surface acting + deep acting.

**Table 5 ijerph-17-04314-t005:** Descriptive statistics and correlations for all measures.

Individual Level Variables’ Descriptive Statistics and Correlations Analysis						
	Mean	Standard Deviation	1	2	3	4
1. Individual service orientation	5.334	1.056	—	—	—	—
2. Individual social capital	5.167	0.976	0.597 **	—	—	—
3. Surface acting	3.139	0.884	0.015	-0.11	—	—
4.Deep acting	3.676	0.781	0.424 **	0.534 **	0.163 **	—
**Organizational Level Variables’ Descriptive Statistics and Correlations Analysis**						
1. Organization Service Orientation	5.070	0.829	—	—	—	—
2. Organizational social capital	5.194	0.471	0.301 **	—	—	—
3. Aggregated surface acting	3.131	0.426	−0.268	−0.130	—	—
4. Aggregated surface acting	3.682	0.310	0.220	0.788 **	0.140	—

** *p* < 0.01.

**Table 6 ijerph-17-04314-t006:** Multilevel antecedent’s examining results of individual social capital.

Dependent Variable	Individual Social Capital	
	M1	M2
Intercept	5.023 ***	1.462 **
Control variable		
Gender	0.034	0.131 *
Age	0.021	0.010
Marital status	−0.007	0.032
Educational level	0.069	0.072
Income	0.023	0.033
Working years	−0.059	−0.032
Independent variable (Level 1)		
Individual service orientation		0.501 ***
Independent variable (Level 2)		
Organization service orientation		0.141 **
R²	74.76%	43.83%

** p* < 0.05; ** *p* < 0.01; *** *p* < 0.001.

**Table 7 ijerph-17-04314-t007:** Examining the impact of individual social capital on emotional labor.

Dependent Variable	Emotional Labor (Surface Acting, Deep Acting)	
	M1 (Surface Acting)	M2 (Deep Acting)
Intercept	3.132 ***	3.773 ***
Level1 control variables		
Gender	−0.060	−0.037
Age	−0.036	0.029
Marital status	0.016	0.024
Education level	−0.039	−0.003
Income	0.042	−0.012
Working years	0.061	−0.058
Independent variable (Level 1)		
Individual social capital	0.010	0.372 ***

*** *p* < 0.001. In M1 (Table 7), the regression coefficient of individual social capital on surface acting was 0.010 (*p* > 0.05), thereby not supporting H3. In M2 (Table 7), the regression coefficient of individual social capital on deep acting was 0.372 (*p* < 0.001), thereby supporting H4.

**Table 8 ijerph-17-04314-t008:** The mediating role of individual social capital (the first conditional test).

Dependent Variable	Emotional Labor (Surface Acting, Deep Acting)			
	M1 (Surface Acting)		M2 (Deep Acting)	
Intercept	3.003 ***	3.883 ***	2.023 ***	1.507 ***
Control variables (Level 1)				
Gender	−0.052	−0.046	0.033	0.024
Age	−0.021	−0.021	0.034	0.036
Marital status	0.017	0.020	0.014	0.015
Education level	−0.049	−0.057	0.008	0.023
Income	0.050	0.051	0.011	0.010
Working years	0.052	0.047	−0.056	−0.055
Independent variable (Level 1)				
Individual service orientation	0.020		0.300 ***	
Independent variable (Level 2)				
Organization Service Orientation		−0.147		0.094 **
R²	56.65%	56.61%	41.11%	41.19%

** *p* < 0.01; *** *p* < 0.001.

**Table 9 ijerph-17-04314-t009:** The mediating role of individual social capital (the fourth condition test).

Dependent Variable	Emotional Labor (Surface Acting, Deep Acting)	
	M1 (Surface Acting)	M2 (Deep Acting)
Intercept	3.913 ***	1.011 ***
Control variables (Level 1)		
Gender	−0.030	−0.025
Age	−0.036	0.027
Marital status	0.020	0.019
Education level	−0.068	−0.002
Income	0.042	−0.003
Working years	0.056	−0.027
Independent variable (Level 1)		
Individual service orientation	−0.004	0.126 ***
Individual social capital	0.029	0.321 ***
Independent variable (Level 2)		
Organization Service Orientation	−0.160 *	0.065 *
R²	54.29%	35.77%

** p* < 0.05; *** *p* < 0.001.

**Table 10 ijerph-17-04314-t010:** Regression analysis of organizational service orientation and organizational social capital.

Independent Variable	Dependent Variable	Standard Coefficient	T	Sig.	Collinearity Diagnosis	
		Beta			Tolerance	VIF
Organizational service orientation	Organizational social capital	0.247 *	1.372	0.181	1.000	1.000

** p* < 0.05; In Table 10, the regression coefficient of organizational service orientation and organizational social capital was 0.247 (*p* < 0.05), thereby supporting H9.

**Table 11 ijerph-17-04314-t011:** Regression analysis of organizational social capital and aggregated emotional labor.

Independent Variable	Dependent Variable	Standard Coefficient	T	Sig.	Collinearity Diagnosis	
		Beta			Tolerance	VIF
Organizational social capital	Aggregated surface acting	−0.130	−0.708	0.485	1.000	1.000
Organizational social capital	Aggregated deep acting	0.588 ***	6.891	0.000	1.000	1.000

*** *p* < 0.001. The regression coefficient of organizational social capital and aggregated surface acting was −0.130 (*p* > 0.05), thereby not supporting H10. In addition, the regression coefficient of organizational social capital and aggregated deep acting was 0.588 (*p* < 0.001), thereby supporting H11.

**Table 12 ijerph-17-04314-t012:** Research hypothesis test results.

Hypothesis	Test Result
H1 Individual service orientation has positive effect on individual social capital	Supported
H2 Organizational service orientation has positive effect on individual social capital	Supported
H3 Individual social capital has negative effect on surface acting	Not supported
H4 Individual social capital has positive effect on deep acting	Supported
H5 Individual social capital has mediating effect between individual service orientation and surface acting	Not supported
H6 Individual social capital has mediating effect between individual service orientation and deep acting	Supported
H7 Individual social capital has mediating effect between organizational service orientation and surface acting	Not supported
H8 Individual social capital has mediating effect between organizational service orientation and deep acting	Supported
H9 Organizational service orientation has positive effect on organizational social capital	Supported
H10 Organizational social capital has negative effect on aggregated surface acting	Not supported
H11 Organizational social capital has positive effect on aggregated deep acting	Supported

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
