# Peer review of "The Multilevel Mechanism of Multifoci Service Orientation on Emotional Labor: Based on the Chinese Hospitality Industry"

_ijerph, 2020, doi:10.3390/ijerph17124314_

Round 1

Reviewer 1 Report

I want to thank you for the opportunity to review this manuscript. The time spent creating and shipping it is greatly appreciated. IMHO, it offers interesting results that can benefit the scientific community as well as professionals in this field. However, currently the manuscript presents some problems that must be taken into account and repaired. Below I present my recommendations separated by sections. Hopefully they will be useful:

Abstract
They must expand the information on the characteristics of the sample.

Introduction
The introduction is correct, however, I recommend that you introduce some studies published in the last two years so that the study topic is updated as much as possible.

Discussion
In the discussion section, a comparison is made of the results found with those of other authors previously mentioned in the introduction. There are studies that have been cited in the discussion, but not in the introduction. For this reason, I suggest that the discussion be revised, especially considering the appropriate modifications that are carried out in the introduction.

Conclusions
I think the conclusion can be expanded to include and / or give examples of how this information can be useful to the population and the health field. The authors should make explicit reference to the practical application of the specific results (this is one of the strengths of the manuscript), paying special attention to the possibilities offered by the data for the design of controls (prevention).

Author Response

Dear reviewer 1,

Thank you for giving us the opportunity to revise our paper. Indeed, we put lots of effort into this paper. We rewrote the introduction, literature view, hypnosis development, conclusions and implications in the revised manuscript. Also, we are glad to accept all your suggestions and carefully revise each question one by one.

The detail of our response, please see the attachment.

We hope that every issue would be addressed and meet with approval. Finally, we thank you once again for your insightful and valuable comments and for your time spent on the manuscript. Because of your input, our revised manuscript would be greatly improved. We greatly appreciate your kind and constructive comments.

Yours truly,

The Authors

Reviewer 2 Report

The author does not explain why their study is needed and what adds to the previous literature.

The author introduces constructs and concepts without explaining them, such as "social capital""multifoci service orientation", "multilevel theoretical perspective", "individual service orientation", "multifoci emotional labor, multifoci social capital".

More than that, the reasoning behind the hypotheses and  how they are backed by the previous literature are too poorly elaborated. Author puts together information, but does not explain how they are logically linked. For instance, line 77-79 "However, as service warrants high interaction among service employees and needs customers high participation, employees with higher service orientation are more likely to foster higher social capital".

Other sentences look like truism, such as lines 166-167 "When organizational service orientation is high, service organizations focus on providing high-quality services to customers". or lines 110-112 "Precisely, when a servic employee’s social capital is high, he/she can build a trust relationship with his/her colleagues and customers".

The communication and the English is rather confused and confusing;  as as consequence, it is difficult to understand what the author is really willing to address

The literature review is not sufficiently developed, this is also a problem when the reader tries to follow the development of the ideas through the manuscript

The phases of the study are not clearly described; it is rather difficult to understand what happened, what was administered in each phase, and why.

For al the above reason it is difficult to assess th the conclusions

I suggest to re-write the whole manuscript, improving the readability (both in terms of English and reasoning), as well as  the literature review

Author Response

Dear reviewer 2,

Thank you for giving us the opportunity to revise our paper. Indeed, we put lots of effort into this paper. We rewrote the introduction, literature view, hypnosis development, conclusions and implications in the revised manuscript. Also, we are glad to accept all your suggestions and carefully revise each question one by one.

The detail of our response, please see the attachment.

We hope that every issue would be addressed and meet with approval. Finally, we thank you once again for your insightful and valuable comments and for your time spent on the manuscript. Because of your input, our revised manuscript would be greatly improved. We greatly appreciate your kind and constructive comments.

Yours truly,

The Authors

Reviewer 3 Report

Dear Authors,

Thanks for the opportunity to review the manuscript titled: “The multilevel mechanism of multifoci service  orientation on emotional labor: Based on the Chinese  hospitality industry".

The manuscript addresses an issue of the multilevel mechanism of multifoci service  orientation on emotional labor from the perspective of social capital theory. The topic itself is interesting and within ta scope of IJERPH.

In my opinion, there are some shortcomings that need to be corrected in order for the paper to be acceptable:

  1. I would recommend extending hypotheses development. In my opinion The literature review must be further elaborated highlighting the relationship of variables in the conceptual framework. Study need more bibliographical references anchoring it into the current scientific background.

  1. I suggest elaborating on social capital theory

Putnam, Robert D., 1993, Making Democracy Work: Civic Traditions in Modern Italy (Princeton University Press, Princeton, NJ).

Putnam, RobertD., 2000, Bowling Alone: The Collapse and Revival of American Community (Simon and Schuster, New York, NY).

Lin, Nan, 2001, Social Capital: A Theory of Social Structure and Action (Cambridge University Press, Cambridge, UK).

  1. I would suggest Authors include some justification regarding a linkage between study variables and control variables (demographic variables) (lines 281-282). Please refer to the recommendation from :

Thomas E. Becker, (2005), Potential Problems in the Statistical Control of Variables in Organizational Research: A Qualitative Analysis with Recommendations, Organizational Research Methods;  Jul 2005; 8, 3, p. 274-289

  1. My main concerns refers to the verification of hypotheses. (Lines 350-361) Authors state that they have tested hypotheses according to Preacher and Hayes (which is commonly accepted method). However, next they claim they have analysed data using correlation. Hayes and Preacher mediation analysis based on linear regression-based path.  This is need to be addressed. (Hayes, Preacher, 2014)

  1. I would advise elaborating the implication further based on your findings. Do not try to generalize. Try to be specific to your research and bring implications.

In my opinion, the manuscript  needs major revision. In conclusion, this paper is good, but some improvements are necessary for the final publication.

Author Response

Dear reviewer 3,

Thank you for giving us the opportunity to revise our paper. Indeed, we put lots of effort into this paper. We rewrote the introduction, literature view, hypnosis development, conclusions and implications in the revised manuscript. Also, we are glad to accept all your suggestions and carefully revise each question one by one.

The detail of our response, please see the attachment.

We hope that every issue would be addressed and meet with approval. Finally, we thank you once again for your insightful and valuable comments and for your time spent on the manuscript. Because of your input, our revised manuscript would be greatly improved. We greatly appreciate your kind and constructive comments.

Yours truly,

The Authors

Round 2

Reviewer 2 Report

The manuscript has significantly improved

Reviewer 3 Report

Dear Authors,

Thank you for including all the remarks and suggestions.

Your manuscript is now ready to be published.

Good luck!

Best Regards.